# Cutaneous Complications of mRNA and AZD1222 COVID-19 Vaccines: A Worldwide Review

**DOI:** 10.3390/microorganisms10030624

**Published:** 2022-03-15

**Authors:** George Kroumpouzos, Maria Eleni Paroikaki, Sara Yumeen, Shashank Bhargava, Eleftherios Mylonakis

**Affiliations:** 1Department of Dermatology, Warren Alpert Medical School at Brown University, Providence, RI 02903, USA; sara_yumeen@brown.edu; 2Department of School of Medicine, Imperial College London, London SW7 2BX, UK; maria.paroikaki18@imperial.ac.uk; 3Department of Dermatology, R.D. Gardi Medical College, Ujjain 456010, India; shashank2811@gmail.com; 4Division of Infectious Diseases, Rhode Island Hospital, Warren Alpert Medical School of Brown University, Providence, RI 02903, USA

**Keywords:** COVID-19 vaccine, mRNA vaccine, rash, skin reaction, delayed hypersensitivity reaction, herpes zoster, urticaria, morbilliform, chilblains

## Abstract

Because of the increasing emergence of cutaneous reactions from COVID-19 vaccines worldwide, we investigated the published reports of these complications. We searched the PubMed, Google Scholar, and Scopus databases and the preprint server bioRxiv for articles on cutaneous complications linked to mRNA-1273 (Moderna), BNT162b2 (Pfizer–BioNTech), and AZD1222 (AstraZeneca–Oxford University) vaccines published until 30 September 2021. Eighty studies describing a total of 1415 reactions were included. Cutaneous reactions were more prevalent in females (81.6%). Delayed large local reactions were the most common complication (40.4%), followed by local injection site reactions (16.5%), zoster (9.5%), and urticarial eruptions (9.0%). Injection site and delayed large local reactions were predominantly caused by the mRNA-1273 vaccine (79.5% and 72.0%, respectively). BNT162b2 vaccination was more closely linked to distant reactions (50.1%) than mRNA-1273 (30.0%). Zoster was the most common distant reaction. Of reactions with adequate information for both vaccine doses, 58.3% occurred after the first dose only, 26.9% after the second dose only, and 14.8% after both doses. Overall, a large spectrum of cutaneous reaction patterns occurred following the COVID-19 vaccination. Most were mild and without long-term health implications. Therefore, the occurrence of such dermatologic complications does not contraindicate subsequent vaccination.

## 1. Introduction

Coronavirus disease 2019 (COVID-19) vaccines are an effective tool in reducing the risk of developing COVID-19 and serious adverse outcomes. Cutaneous complications have been associated with COVID-19 vaccination [1,2]. In a US study, cutaneous adverse effects associated with the first dose of messenger RNA (mRNA) vaccines were reported by 1.9% (95% CI, 1.8–2.1%) of health care employees [3]. Of those, 83% reported no recurrent cutaneous reactions. In a prospective observational study from the UK, first-dose and second-dose skin reactions were observed in only 1.1% and 1.7% of patients, respectively, after BNT162b2 (Pfizer–BioNTech) vaccination [4]. While the cutaneous complications of these vaccines may be reported less frequently, they nonetheless impact public perception regarding vaccine safety. The objective of this systematic review was to assess the dermatologic complications of mRNA-1273 (Moderna; mRNA vaccine), BNT162b2, and AZD1222 (AstraZeneca–Oxford University; adenovirus vector vaccine) vaccination. 

## 2. Methods

### 2.1. Search Strategy

A search in the PubMed, Google Scholar, and Scopus databases and the preprint server bioRxiv was conducted for articles related to cutaneous complications linked to mRNA COVID-19 vaccines. The search strategy included a combination of key terms: (‘COVID-19’ OR ‘SARS-CoV-2’) AND (‘vaccine’ OR ‘vaccination’) AND (‘skin’ OR ‘cutaneous’) AND (‘rash’ OR ‘reaction’ OR ‘eruption’ OR ‘complication’ OR ‘lesion’ OR ‘flare’ OR ‘delayed’ OR ‘urticaria’ OR ‘morbilliform’ OR ‘herpes zoster’ OR ‘chilblains’ OR ‘eczema’ OR ‘psoriasis’ OR ‘vesicular’ OR ‘bullous’). Abstracts of papers published until September 30, 2021, were reviewed. When an abstract was unavailable, we reviewed the text of the article. A manual search of references cited in the selected articles and published reviews was also used to highlight undetected studies. 

### 2.2. Study Selection

The study selection is detailed in the Appendix A. Eligibility assessment was performed independently by two authors (G.K and M.P.). Inclusion criteria were studies published in the English language and reporting cutaneous reactions from COVID-19 vaccines. The exclusion criteria were laboratory cell/animal studies, review/opinion articles, commentaries, consensus papers, editorials, reports missing the vaccine type, studies not focusing on cutaneous reactions, studies with incomplete clinical data (e.g., clinical features and/or course of the eruption), histopathologic studies with inadequate clinical data, and self-reported reactions. We excluded reports of reactions after booster vaccination and those associated with the CoronaVac (SinoVac) vaccine because the number was minimal (insufficient data). Any disagreements in terms of study selection were discussed among the co-authors until a consensus was reached. 

### 2.3. Extraction of Data

We extracted the following data: study design, patient country, vaccine type(s) administered, number of persons vaccinated and the male–female ratio, type(s) of cutaneous reaction(s) observed, total reactions, reactions per vaccine type, number of reactions to first vaccine dose, number of reactions to second vaccine dose, time to onset of reaction after vaccination, time to resolution of reaction, and intervention. 

### 2.4. Quality of Evidence Assessment

Quality rating of the studies was ranked according to the Quality Rating Scheme for Studies and Other Evidence [5] and the Oxford Centre for Evidence-based Medicine for ratings of individual studies [6]. Biases of all included studies were assessed. 

## 3. Results

We summarize the results of 80 studies (1415 reactions; Table 1) of which 4 were registry-based, 1 was a cross-sectional national study, 1 was a retrospective study, 39 were case series, and 35 were case reports. Delayed large local reactions (DLLLs) were the most common complication (40.4%), followed by local injection site reactions (16.5%), zoster (9.5%), urticarial (9.0%), and morbilliform/diffuse erythematous eruptions (6.7%) (Table 2). In a total of 1265 patients, 618 (48.9%) were from Europe (534 from Spain; 42.2%), 592 (46.8%) from USA, 49 (3.9%) from Asia, and 6 (0.5%) from the rest of the World. There was a female predominance (81.6%). Two large studies did not report differences in types of reactions among age groups [1,2]. In these studies, the median age of participants was 44 years (interquartile range, 36–59 years) [1] and the mean age was 50.7 years [2].

Most reactions (55.7%) were associated with mRNA-1273 vaccination (Table 2). Injection site reactions and delayed large local reactions were predominantly caused by the mRNA-1273 vaccine (79.5% and 72.0%, respectively). BNT162b2 vaccination was more closely linked to distant reactions (334/610; 54.8%) than mRNA-1273 (191/610; 31.3%) (Table 2). Of reactions with adequate information for both vaccine doses (*n* = 1361), 58.3% occurred after the first dose only, 26.9% after the second dose only, and 14.8% after both doses. Potential mechanisms underlying cutaneous reactions are summarized in Table 3.

### 3.1. Quality of Evidence Assessment

The rating score of the studies included is shown in Table 1. There were only a small number of registry-based studies and cohorts [1,2,7,18,25,30], and the sample size of some outcomes (cutaneous reactions) in registries/cohorts documenting various outcomes (different types of cutaneous reactions) was small (Table 1). Reporting bias applied, as evidenced by limited data for AZD1222 and the fact that most cutaneous reactions were documented in white persons. Studies performed in health care workers confirmed reporting bias, as healthcare workers are more likely to report their reactions [1,7]. A registry-based study may have included a confirmation bias, as providers are more likely to report cases with severe or rare manifestations [1]. Biases relevant to retrospective observational studies, such as selection and information biases (e.g., short follow-up period; course of the reaction determined mainly based on the patient’s description), also applied.

### 3.2. Local Site Injection Reaction

Local injection site reaction can be immediate (median of 1 day after first dose) or delayed (median of 7 days after first dose) [1]. Immediate site reaction can manifest with edema/erythema and often pain. DLLL, also called the ‘COVID arm’, occurs at or near the vaccination injection site [1,2]. This reaction was more common with mRNA-1273 than BNT162b2 and AZD1222 vaccination (Table 1), and this concurs with previous studies [1,2]. In the study by Català et al., the ‘COVID arm’ was much more common in females (95.4%) and is more likely to be associated with systemic symptoms (64.6%) than other post-vaccination eruptions [2]. It can manifest as a solitary pink patch or plaque associated with erythema, induration, pruritus, and/or tenderness (Figure 1) [7,8,9,10,11,12,13,14,15,16,17]. Severe reactions with lesion sizes of >10 cm have been reported [9]. Five patients (4.9%) in a cohort presented disseminated lesions [7]. Systemic symptoms such as fever, headache, and chills can be present [1,9].

Second-dose DLLLs generally occur more quickly (median of 2–3 days) [1,9,10,11]. Such reactions were fewer in the AAD/ILDS registry but not in a cohort of 103 COVID arm cases associated with BNT162b2 vaccination (54% of reactions) [1,7]. The duration of second dose DLLLs was longer than first dose examples in most studies [1,9,10]. In the American Academy of Dermatology/International League of Dermatologic Societies (AAD/ILDS) registry, the majority of patients who developed a DLLL after both doses of the mRNA-1273 or BNT162b2 vaccines showed a larger reaction after the second dose [1]. 

Histopathology of DLLLs showed perivascular lymphocytic infiltrates with eosinophils and scattered mast cells consistent with a delayed T-cell mediated hypersensitivity reaction [9,89]. The presence of prominently dilated vessels with edematous endothelial layers was a consistent feature [13]. ‘COVID arm’ typically resolves within one week of treatment with topical corticosteroid, oral antihistamines, and symptomatic therapy. Many cases have been treated with expectant management [2,11].

### 3.3. Urticaria 

Urticaria can develop as an immediate hypersensitivity reaction, defined by the Centers of Disease Control and Prevention (CDC) as an onset within 4 h after injection, or can occur ≥4 h after injection. The former is a potential contraindication to the second dose. One hundred twenty-eight cases of urticarial eruption were reported in 12 studies [1,2,16,17,18,19,20,21,22,23,24,25], of which 57 (44.5%) were after BNT162b2, 47 (36.7%) after mRNA-1273, and 24 (18.8%) after AZD1222 vaccination (Figure 2). Of these cases, 11 were labeled by the CDC as part of an anaphylaxis reaction and submitted to the Vaccine Adverse Event Reporting System (VAERS) [18]. Several cases of urticaria were associated with angioedema [2]. However, in the AAD/ILDS registry, none of the 40 urticarial reactions were classified as immediate hypersensitivity reactions [1].

Anaphylaxis has developed within 150 min post-COVID-19 vaccination. It is uncommon; of 1,893,360 individuals who received the first BNT162b2 vaccine dose, the Food and Drug Administration (FDA) reported 21 patients with an anaphylactic reaction [18]. Of those, 19 were female, 2 were male, and 17 had a history of allergies or allergic reactions. The reaction occurred at a median of 13 min post-vaccination. Of 4,041,396 individuals that received the mRNA-1273 vaccine, 10 females experienced anaphylaxis after the first dose [25]. Nine of 10 patients had a history of atopic disease, and anaphylaxis occurred a median of 7.5 min post-vaccination. All patients were treated with an emergency intramuscular or subcutaneous epinephrine injection [18,25]. Mechanisms of anaphylactic reaction are shown in Table 3 [85].

### 3.4. Mobilliform Eruption

Twelve studies detailed 95 morbilliform/maculopapular eruptions, of which 53 (55.3%) were after BNT162b2, 31 (33.0%) after mRNA-1273, and 11 (11.7%) after AZD1222 vaccination (Table 1; Figure 3) [1,2,17,18,22,23,24,25,26,27,28,29]. In the AAD/ILDS registry, such eruptions started at a median of 3 days after the first dose and two days after the second dose [1]. In the study by Català et al., such eruptions started at a mean of 4 days after vaccination and lasted a mean of 10.3 days [2]. The authors indicated that morbilliform eruption was the earliest cutaneous reaction pattern that appeared. Half of the morbilliform eruptions were classified as grade 3 (severe) or grade 4 (very severe) in the study. In a series of five patients with morbilliform eruption, three patients had history of atopic dermatitis and one of angioedema [26]. Itching was reported in most patients [2,26]. 

Among the cases that were not associated with anaphylaxis, most eruptions developed within 2 to 3 days post-vaccination and resolved within a week. A generalized eruption (>30% of body surface area covered) in one participant that received the BNT162b2 vaccine persisted for more than one month [28]. The patient had no significant past medical history or drug allergy. Histopathology showed lymphocytic perivascular infiltrates consistent with maculopapular eruption. A laboratory investigation showed increased liver enzymes and the second vaccine dose was not provided. Tihy et al. indicated that morbilliform eruptions shared histopathologic similarities with drug eruption [17]. Ohsawa and colleagues demonstrated similarities between the immunohistochemical features of morbilliform eruption in one case and those found in COVID-19-associated skin lesions [86]. When treatment is required, morbilliform eruptions respond to topical/systemic corticosteroids and oral antihistamines.

### 3.5. Varicella Zoster Virus (VZV) and Herpes Simplex Virus (HSV) Reactivation

There have been 12 reports of VZV or HSV reactivation after vaccination [1,2,30,31,32,33,34,35,36,37,38,39]. Fathy et al. published a series of 40 cases of VZV (*n* = 35) or HSV reactivation (*n* = 5) after BNT162b2 or mRNA-1273 vaccination [30]. VZV was reported at a median of 7 days and HSV reactivation at a median of 13 days following vaccination. The median onset of symptoms was 7 days post-vaccination for VZV reactivation and 13 days for HSV reactivation. The median duration of symptoms was 7 days for both groups. Two cases of zoster ophthalmicus were reported [32,33]. In several cases, healthy young individuals developed VZV after BNT162b2 or mRNA-1273 vaccination [31,37]. 

### 3.6. Pityriasis Rosea-like Eruption

Pityriasis rosea is a complication that appeared within 22 days and resolved within 12 weeks of vaccination [1,2,16,17,21,23,40,41,42,43,44,45,46,47,48,49,50]. It was associated with BNT162b2 vaccination in most patients (71.9%; 41 of 57 eruptions, Table 1). In a series of 14 patients, the median onset was 14 days after the first vaccine dose and 9 days after the second [40]. In a cross-sectional study, pityriasis rosea-like eruption was the longest-lasting cutaneous reaction pattern [2]. These authors report a similar case (Figure 4). McMahon et al. proposed that the most common histopathologic reaction pattern for pityriasis rosea and other cutaneous reactions was spongiotic dermatitis, which clinically ranged from robust papules with an overlying crust to pink papules with fine scales [89].

### 3.7. Pernio, Chilblains, and Purpura

Pernio-like lesions and purpuric eruptions have been reported post-COVID-19 vaccination (Figure 5). Of the 41 cases described in 11 observational studies [1,2,21,23,24,27,51,52,53,54,55,56], 25 (61.0%) were associated with BNT162b2, 9 (22.0%) with ADZ1222, and 7 (17.1%) with mRNA-1273 vaccination (Table 1). In a series of 16 patients, purpuric manifestations appeared at a mean of 7.6 days after COVID-19 vaccination and lasted a mean of 15.7 days [2]. Pernio and purpuric manifestations were delayed-type reactions, as they typically happened days following exposure.

### 3.8. Delayed Inflammatory Reaction (DIR) to Dermal Hyaluronic Acid Filler

DIR to hyaluronic acid dermal filler presents clinically as edema with inflammatory, erythematous nodules at the site of prior dermal filler injections. The AAD/ILDS registry reported one DIR after BNT162b2 and eight after mRNA-1273 vaccination [1]. Munavalli and colleagues reported three DIRs after BNT162b2 and four after mRNA-1273 vaccination [57,58]. The reactions occurred within 10 days after vaccination. Marked improvements were noted within 5 days of lisinopril 5–10 mg administration in all patients. In a patient who developed DIR after the first mRNA-1273 dose, preventive lisinopril treatment was successful before the second dose [58]. Angiotensin-converting enzyme 2 inhibitors (ACE-I), such as lisinopril, can block ACE2 receptor targeting by the SARS-CoV-2 spike protein that releases a proinflammatory cascade. This observation may explain the efficacy of lisinopril treatment in the above DIRs. A case was treated with hyaluronidase injection [59]. The American Society for Dermatologic Surgery released guidance in which it was outlined that patients with dermal fillers do not have any contraindication to receiving any COVID-19 vaccine, and that those who already received the vaccine remain candidates for the future receipt of dermal filler [90]. 

### 3.9. Unusual Reactions

#### 3.9.1. Papulovesicular Lesions

In a series of 26 patients, the average time to onset was 6.4 days, and lesions lasted an average of 19.3 days [2]. 

#### 3.9.2. Vesiculobullous Lesions

Vesiculobullous lesions have been reported [1,23]. Some cases showed features of bullous pemphigoid [50,60] or linear IgA dermatosis [60].

#### 3.9.3. Erythromelalgia

Of 14 cases of erythromelalgia, 11 (79%) were associated with the mRNA-1273 vaccine [1].

#### 3.9.4. Eczematous Eruption

A pruritic generalized eczematous eruption was described in three patients within 14 days post-BNT162b2 vaccination [22,61]. Two patients had a history of atopic dermatitis and another dyshidrotic eczema. Cases of localized eczematous dermatitis and hematogenous contact dermatitis have been reported [23,24].

#### 3.9.5. Other Eruptions

Three of five erythema multiforme cases were associated with the mRNA-1273 vaccine [1,62,63]. New onset of prurigo nodularis [17], radiation recall dermatitis [64], symmetrical drug-related intertriginous and flexural exanthema (SDRIFE)-like eruption [65], Stevens-Johnson syndrome/toxic epidermal necrolysis [66,67,68], Sweet’s syndrome [69,70], vitiligo [71], vasculitis [50,72,73], livedo racemosa [27], fixed drug eruption [27], pityriasis rubra pilaris-like eruption [74], and facial pustular neutrophilic eruption [75] have been reported post-vaccination. All SJS and Sweet’s syndrome cases were managed successfully. Lastly, the inflammation of bacillus Calmette-Guérin (BCG) scars developed within 30 h of BNT162b2 or mRNA-1273 vaccination and were resolved within 4 days [76].

#### 3.9.6. Exacerbation of Pre-Existing Skin Condition

Psoriasis vulgaris [23,77,78], generalized pustular psoriasis [79], guttate psoriasis [80], palmoplantar psoriasis [81], and cutaneous lupus [23,82] have flared or developed after vaccination. These authors report a similar case (Figure 6). Atopic dermatitis [23], Darier’s disease [83], and lichen planus [84] can also flare post-vaccination.

## 4. Discussion

DLLLs were the most common post-vaccination skin complication, followed by local injection site reactions, urticarial eruptions, zoster, and morbilliform eruptions. Most local reactions were associated with the mRNA-1273 vaccine and most distant reactions with BNT162b2. Zoster was the most common distant reaction. To our knowledge, this finding has not been reported. There is considerable geographic variation because most participants in the studies included were from Europe and the USA. Most patients (81.6%) that developed cutaneous reactions were female [1,2,7]. Female predominance was observed not only in US studies that included the health care workforce (consisting of 76% females [91]), which might reflect a reporting bias [1], but in European studies as well [2,7]. Some authors propose that women’s immune systems may be more reactive to coronavirus proteins, leading to a lower susceptibility to the disease and a higher reactogenicity to vaccines [2].

Most reactions were effectively managed with minimal to no long-term morbidity, and the completion of the vaccination course was recommended [1]. Anaphylactic reactions are rare with COVID-19 vaccines [18], and the incidence has been similar to what is noted with other virus-based vaccines [92]. Fatalities were not reported. CDC recommends that vaccination be contraindicated in patients who have had a severe or immediate allergic reaction to the COVID-vaccine or any of its components and that clinicians consider a referral to an allergist-immunologist in such cases [18]. As most people that experienced anaphylaxis had allergy histories [18,25,93], it is very important that clinicians screen for a history of anaphylaxis or angioedema or a proclivity to allergic reactions, e.g., a history of atopy or allergic reactions to vaccine components. Individuals with histories of allergic reactions to one or several of the COVID-19 vaccine ingredients should not receive vaccination [94]. Receiving a different COVID-19 vaccine for the second dose is appropriate for patients with a proclivity to allergy experiencing first-dose reactions. Studies have shown that heterologous prime-boost vaccines are effective and may provide higher immunogenicity than using the same vaccine for booster doses [95]. 

Some patients experienced reactions to mRNA vaccines, such as pernio/chilblains and erythromelalgia, which mimicked COVID-19 infection [1]. This finding suggests that the vaccine replicates the host immune response to the virus, and some components of such cutaneous reactions result from an immune response to the virus rather than direct viral effects. It is important that clinicians distinguish cutaneous reactions to vaccines from signs of COVID-19 occurring post-vaccination. However, in some cases, the development of COVID-19 after immunization cannot be excluded as a plausible cause of cutaneous reactions. Still, available data suggest that prior COVID-19 does not predetermine cutaneous reactions, or reactions of a greater severity, after vaccination [2]. 

This review has several limitations. The search for articles was restricted to those written in English. Many of the included studies were case reports and studies with small sample sizes that may confer publication bias. Additionally, most studies included participants from the USA and Europe, and there is a lack of data from other parts of the globe. Also, there are limited data for the AZD1222 vaccine. The above may reflect underreporting and limit the generalizability of the results. The short duration of participant selection, including the follow-up period, in large studies is an additional limitation because providers entered data at one point in time [1] and/or the study was conducted within a short period of time [2,7]. Lastly, most vaccine reactions were documented in white individuals, and this raises concerns about disparities in vaccine access, health care access after experiencing an adverse effect, the differential likelihood of reporting to registries, and/or the recognition of such reactions in patients of color [1].

Dermatologists should contribute to the improved documentation of cutaneous reactions and safety monitoring by reporting their observations to VAERS. Appropriate patient counseling regarding cutaneous reactions to COVID-19 vaccines is crucial and prevents generating concerns disproportionate to potential complications. General practitioners should be aware of such reactions and can play an important role in patient counseling. Lastly, the appropriate identification and management of vaccine reactions often requires a multispecialty approach involving dermatology, allergy, and infectious disease specialists [96].

## Figures and Tables

**Figure 1 microorganisms-10-00624-f001:**
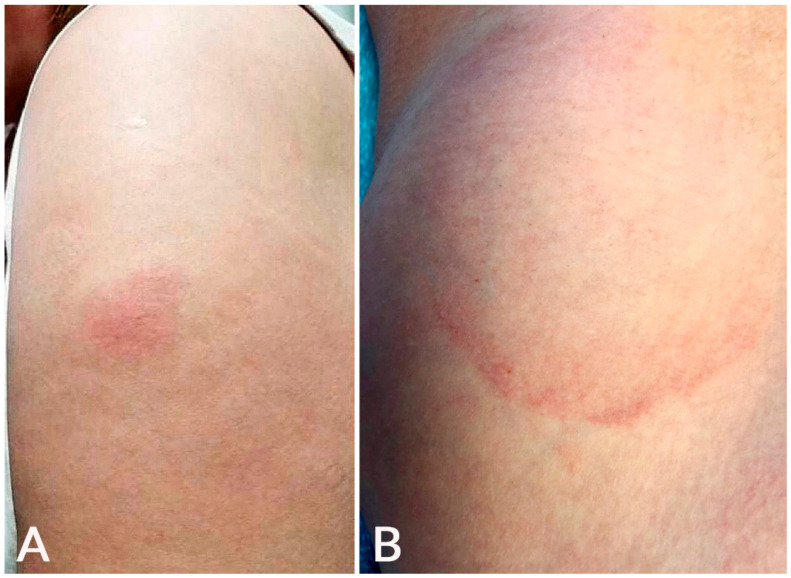
Local injection site reaction: An erythematous plaque developed at the injection site on the left arm 1 day after BNT162b2 vaccination (**A**); delayed large, local reaction: an erythematous, annular, mildly tender plaque developed 6 days after mRNA-1273 vaccination at the injection site on the left arm (**B**).

**Figure 2 microorganisms-10-00624-f002:**
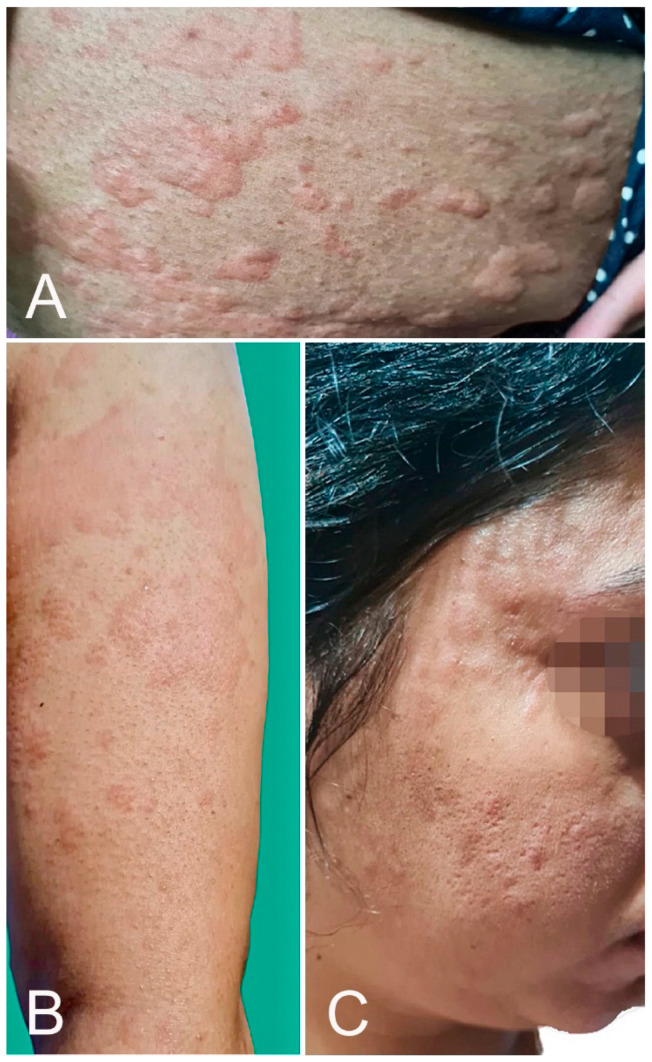
Urticarial eruption: wheals over the upper limb (**A**), trunk (**B**), and face (**C**) developed 2 h after AZD1222 vaccination.

**Figure 3 microorganisms-10-00624-f003:**
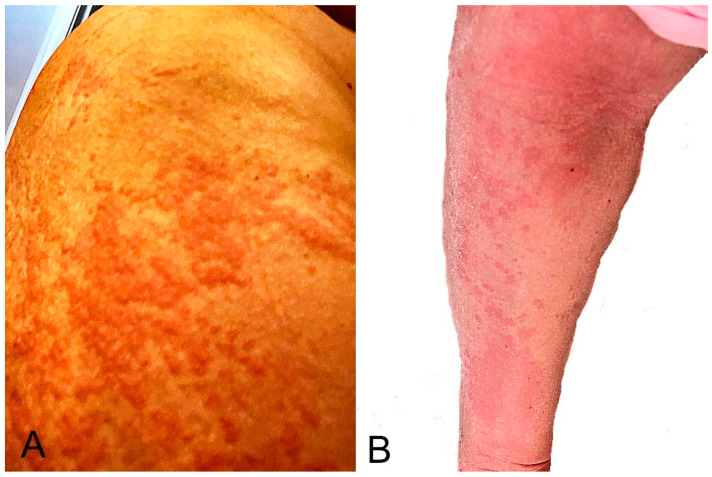
Morbilliform eruption: it started around the vaccination site on the right upper arm (**A**) 2 days after the second mRNA-1273 dose and became generalized. Several areas showed maculopapular lesions with desquamation (**B**). The eruption resolved with a 5-day course of oral prednisone.

**Figure 4 microorganisms-10-00624-f004:**
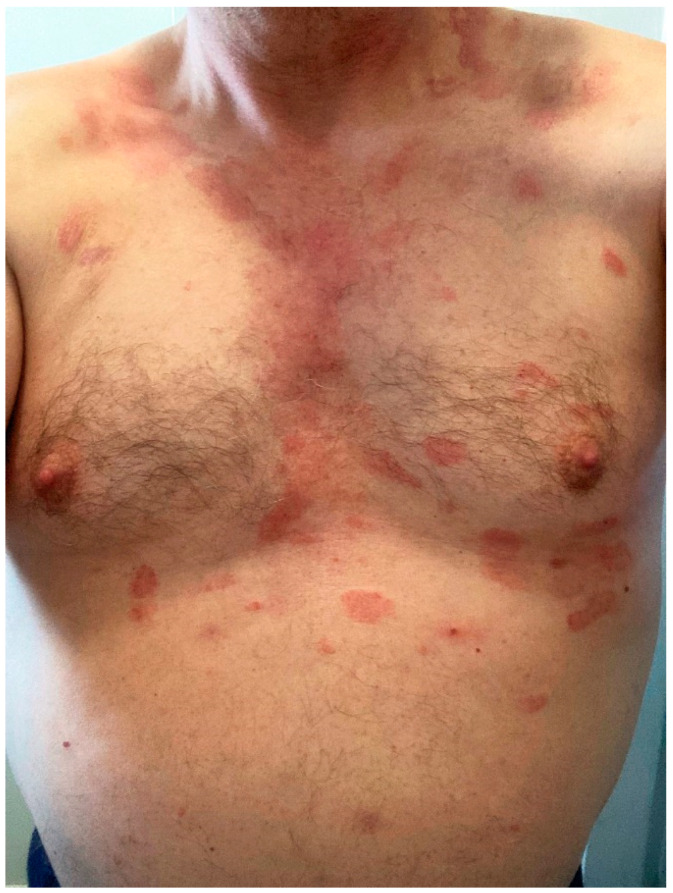
Pityriasis rosea-like eruption: multiple scaly, pink, or red patches developed 20 days after mRNA-1273 vaccination. Histopathology showed features of pityriasis rosea.

**Figure 5 microorganisms-10-00624-f005:**
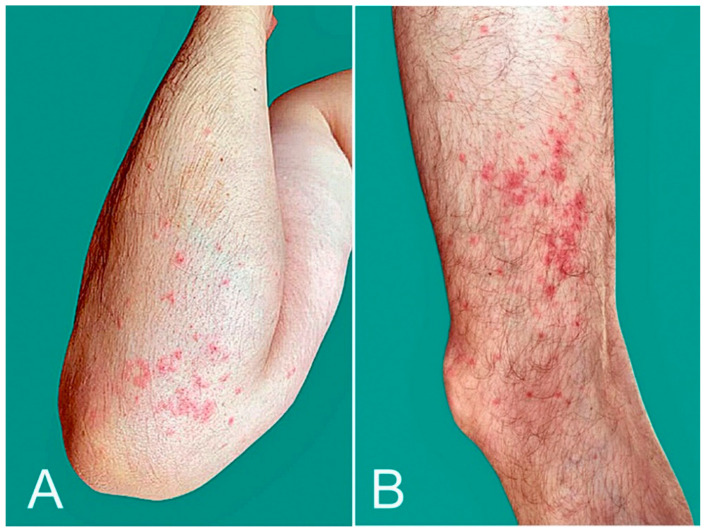
Purpura: palpable purpuric papules over upper (**A**) and lower limbs (**B**) developed 10 days after AZD1222 vaccination.

**Figure 6 microorganisms-10-00624-f006:**
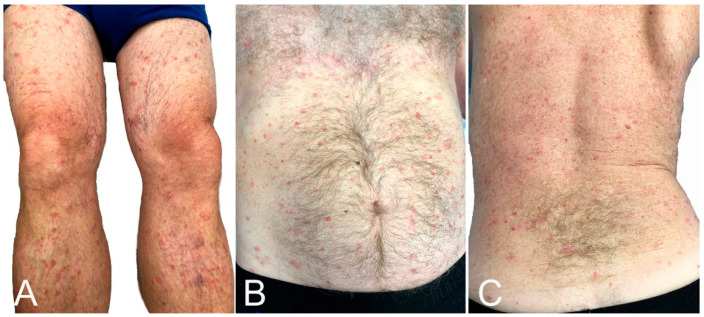
Psoriasis flare: lesions were limited to legs (**A**), but became generalized, affecting the thighs, trunk (**B**,**C**) and upper extremities 5 days after BNT162b2 vaccination.

**Table 1 microorganisms-10-00624-t001:** Cutaneous reactions to mRNA and AZD1222 COVID-19 vaccines reported in the nontrial literature.

Article Reference; Patient Region	Study Design	Rating Score *	Vaccine (Number of Persons); Sex	Cutaneous Reaction	Total Reactions	Reactions to Dose 1	Reactions to Dose 2	Time to Onset after Vaccination (Median)	Time to Resolution (Median)	Intervention
	Local site injection reaction									
McMahon; USA [1]	Registry-based study	3 3	mRNA-1273 (343), BNT16b2 (71); 40 M: 374 F	DLLR local injection site reaction	218 (206 mRNA-1273) 232 (186 mRNA-1273)	180 (175 mRNA-1273) 151 (143 mRNA-1273)	38 (30 mRNA-1273); 11 recurrent (mRNA-1273) 81 (71 mRNA-1273); 21 recurrent (20 mRNA-1273)	**Dose 1:** 7 d **Dose 2:** 2 d **Dose 1:** 1 d **Dose 2:** 1 d	**Dose 1:** 4 d **Dose 2:** 3 d **Dose 1:** 4 d **Dose 2:** 3 d	TCS, OAH, analgesics, ice, antibiotics
Català; Spain [2]	Cross-sectional national study	3	BNT16b2 (163), mRNA-1273 (147), AZD1222 (95); 80 M: 325 F	DLLR	130 (91 mRNA-1273, 23 BNT16b2, 16 AZD1222)	85	45	4.9 d (mean)	7.4 d (mean)	93 rashes: topical/ systemic CS, OAH, paracetamol, NSAIDS, oral antibiotics
Fernandez-Nieto; Spain [7]	Retrospective study	3	BNT16b2 (103); 12 M: 91 F	DLLR	103	49	54; 16 recurrent	NR	<8 h: 23; 8–24 h: 27; 48–72 h: 38; >72 h: 14	NR
Guerrero; Spain [8]	Case series	4	mRNA-1273 (13), BNT16b2 (1); 14 F	DLLR	22	13 (12 mRNA-1273)	9 (mRNA-1273); 8 recurrent	**Dose 1:** 6 d **Dose 2:** 1 d	**Dose 1:** 5 d **Dose 2:** 3 d	TCS (3 patients), OAH (1 patient)
Blumenthal; USA [9]	Case series	4	mRNA-1273 (12); 2 M: 10 F	DLLR	20	12	8; recurrent	**Dose 1:** 8 d **Dose 2:** 2 d	**Dose 1:** 6 d **Dose 2**: 2.5 d	Ice packs, TCS/OCS, OAH, antibiotics
Johnston; USA [10]	Case series	4	mRNA-1273 (16); 3 M: 13 F	Delayed localized hypersensitivity reaction	27	15	12; 11 recurrent	**Dose 1:** 7 d **Dose 2:** 2 d	**Dose 1:** 5 d **Dose 2:** 3 d	TCS, OAH, cool compress, cephalexin
Jacobson; USA [11]	Case series	4	mRNA-1273 (14); 14 F	Delayed injection site reaction	19	13	6; 5 recurrent	**Dose 1:** 7 d **Dose 2:** 2 d	**Dose 1:** 4 d **Dose 2:** 4 d	OAH, low-potency TCS
Ramos; USA [12]	Case series	4	mRNA-1273 (11), BNT16b2 (1); NR	DLLR	12	11 (mRNA-1273)	1 (BNT16b2)	5–11 d (mean, 7 d)	3–8 d (mean, 5 d)	TCS, OAH, ice, analgesics
Hoff; Germany [13]	Case series	4	mRNA-1273 (11); 2 M: 9 F	Delayed skin reaction	11	8	3	**Dose 1:** 7 d **Dose 2:** 3 d	With Rx: 1–2 d Without Rx: 2–4 d	4 patients: TCS, OAH
Wei; USA [14]	Case series	4	mRNA-1273 (4); 4 F	DLLR	4	4	NR	7–10 d	2–4 d	TCS, OAH
Shin; Korea [15]	Case series	4	AZD1222 (4); 4 F	DLLR	4	4	NR	10 d	4 d	NSAIDS, ice packs, SCS
Choi; Singapore [16]	Case series	5	BNT16b2 (1); 1 F	Local injection site reaction	2	1	1	**Dose 1**: 1 d **Dose 2**: NR	NR	NR
Tihy; Switzerland [17]	Case series	5	mRNA-1273 (1); 1 F	DLLR	1	0	1	5 d	<2 wks	NR
	**Urticaria**									
Català; Spain [2]	Cross-sectional national study	3	BNT16b2 (163) mRNA-1273 (147) AZD1222 (95); 80 M: 325	Urticaria and/or angioedema	59 (15 mRNA-1273, 24 BNT16b2, 20 AZD1222)	35	24	4.9 d (mean)	7.5 d (mean)	OAH, TCS, SCS, antibiotics, paracetamol, NSAIDS, epinephrine injection (1 case)
McMahon; USA [1]	Registry-based study	3	mRNA-1273 (343), BNT16b2 (71); 40 M: 374 F	Urticaria	40 (23 mRNA-1273)	25	15; 4 recurrent (3 mRNA-1273)	**Dose 1:** 3 d **Dose 2:** 2 d	**Dose 1:** 5 d **Dose 2:** 3 d	TCS, OAH, analgesics, antibiotics
Team CC-R, FDA; USA [18]	Registry-based study	3	BNT16b2 (10); 1 M: 9 F	Generalized urticaria, anaphylaxis	10	10	NR	5–54 min	NR	Epinephrine injection
Sidlow; USA [19]	Case series	4	mRNA-1273 (6); 1 M: 5 F	Urticarial dermatitis	6	5	1; recurrent	≤3 d	≤17 d	OAH
Kelso; USA [20]	Case series	5	mRNA-1273 (1); 1 F	Urticaria	1	1	No dose 2	6 min	NR	OCS, OAH, epinephrine injection
Yu; Philippines [21]	Case series	4	AZD1222 (3); 3 F	Angioedema (2), urticaria (1)	3	3	NR	Angioedema: 3 h, 3 d Urticaria: 15 min	Angioedema: 3 d, 4 d Urticaria: 14 d	OAH, OCS, epinephrine (angioedema case), TCS, i.m. diphenhydramine,
Corbeddu; Italy [22]	Case series	4	BNT16b2 (2); 2 M	Urticaria	2	2	0	1 h, 2 d	2–3 d	None
Choi; Singapore [16]	Case series	4	BNT16b2 (2); 2 F	Urticaria	2	2	0	**Dose 1**: 17 d, a few d	Dose 1: 2 wks, 6 wks	OAH
Niebel; Germany [23]	Case series	4	BNT16b2 (1), AZD1222 (1); 2 F	Urticaria, generalized hives	2	2	0	1 d, 2 d	NR	OAH
Holmes; USA [24]	Case series	5	mRNA-1273 (1); 1 F	Urticaria, angioedema	1	1	No dose 2	≤2 d	3 d	OAH, baking soda baths
Tihy; Switzerland [17]	Case report	5	BNT16b2 (1); 1 M	Urticarial plaques and papules	1	0	1	21 d	>24 h	NR
Team CC-R, FDA; USA [25]	Registry-based study	5	mRNA-1273 (1); 1 F	Generalized urticarial rash, anaphylaxis	1	1	NR	11 min	NR	Epinephrine injection
	**Morbiliform/diffuse erythematous eruption**									
Català; Spain [2]	Cross-sectional national study	3	BNT16b2 (163), mRNA-1273 (147), AZD1222 (95); 80 M: 325 F	Morbilliform rash	36 (6 mRNA-1273, 19 BNT16b2, 11 AZD1222)	25	11	4 d (mean)	10.3 d (mean)	OAH, TCS, OCS
McMahon; USA [1]	Registry-based study	3	mRNA-1273 (343), BNT16b2 (71); 40 M: 374 F	Morbilliform rash	27 (18 mRNA-1273)	6 BNT16b2 11 mRNA-1273	3 BNT16b2 7 mRNA-1273	**Dose 1:** 3 d **Dose 2:** 2 d	**Dose 1:** 4.5 d **Dose 2:** 2.5 d	TCS, OAH, analgesics, antibiotics
Team CC-R, FDA; USA [18]	Registry-based study	4	BNT16b2 (7); 1 M: 6 F	Diffuse erythematous rash, anaphylaxis	7	7	NR	2–25 min	NR	Epinephrine injection
Tihy; Switzerland [17]	Case series	4	BNT16b2 (4), mRNA-1273 (2); 3 M: 3 F	Morbilliform rash (1); erythematous rash (5)	6 (4 BNT16b2)	1 (BNT16b2)	5 (3 BNT16b2)	**Dose 1**: 8 d **Dose 2**: 13 d (range, 2–16 d)	≤2 wks	
Team CC-R, FDA; USA [25]	Registry-based study	4	mRNA-1273 (4); 4 F	Diffuse erythematous rash, anaphylaxis	4	4	NR	5–45 min	NR	Epinephrine injection
Peigottu; Italy [26]	Case series	4	BNT16b2 (5); 1 M: 4 F	Maculopapular rash	5	1	4	**Dose 1:** 24 h **Dose 2:** 28 h	NR	OAH ± short course OCS
Corbeddu; Italy [22]	Case series	4	BNT16b2 (3); 1 M: 2 F	Morbilliform rash	3	0	3	5 h, 48 h, 3 d	2–3 d	None
Annabi; France [27]	Case series	4	BNT16b2 (2); 2 F	Morbilliform rash	2	2	0	7 d, 8 d	8 d, 15 d	Case 1: none Case 2: TCS
Holmes; USA [24]	Case report	5	mRNA-1273 (1); 1 F	Morbilliform rash	1	0	1	1 d	≤23 d	OCS, OAH, TCS
Ackerman; France [28]	Case report	5	BNT16b2 (1); 1 M	Maculopapular rash	1	1	No dose 2	3 h	1 mo	TCS
Jedlowski; USA [29]	Case report	5	BNT16b2 (1); 1 M	Morbilliform rash	2	1	1; recurrent	48 h	24 h	None
Niebel; Germany [23]	Case series	5	BNT16b2 (1); 1 F	Generalized erythematous plaques	1	1	No dose 2	10 d	NR	Prednisolone
	**VZV/HSV reactivation**									
Fathy; USA [30]	Registry-based study	3	mRNA-1273 (17), BNT16b2 (23); 12 M: 28 F	Zoster, HSV reactivation	35 zoster (19 BNT16b2), 5 HSV (4 BNT16b2)	27 zoster, 4 HSV	8 zoster, 1 HSV	VZV: 7 d HSV: 13 d	VZV: 7 d HSV: 7 d	Systemic antiviral; VZV cases also gabapentin, acetaminophen, TCS
Català; Spain [2]	Cross-sectional national study	3	BNT16b2 (163) mRNA-1273 (147) AZD1222 (95); 80 M: 325 F	Zoster, HSV reactivation	41 zoster (28 BNT16b2, 6 mRNA-1273, 7 AZD1222); 15 HSV (5 BNT16b2, 4 mRNA-1273, 6 AZD1222)	35 (VZV 26)	21 (VZV 15)	VZV: 6.9 d (mean) HSV: 4.6 d (mean)	VZV: 12.1 d (mean) HSV: 9.3 d (mean)	Systemic antiviral; some VZV patients also NSAIDS, topical antibiotics, paracetamol, OAH
McMahon; USA [1]	Registry-based study	4	mRNA-1273 (343), BNT16b2 (71); 40 M: 374 F	Zoster	10 (5 BNT16b2)	6 (5 mRNA-1273)	4 (BNT16b2)	**Dose 1:** 15 d **Dose 2:** 10 d	**Dose 1:** 6 d **Dose 2:** 8 d	NR
Lee; USA [31]	Case series	4	mRNA-1273 (14), BNT16b2 (6); 10 M: 10 F	Zoster	20 (14 mRNA-1273, 6 BNT16b2)	15 (12 mRNA-1273)	5 (3 BNT16b2)	**Dose 1:** 5 d (range, 3–21 d) **Dose 2:** 5 d (range, 3–38 d)	NR	Valacyclovir, gabapentin (8 patients), topical lidocaine 4% (5 cases), OCS, tramadol, TCS, Terrasil shingles cream
Psichogiou; Greece [32]	Case series	4	BNT16b2 (7); 4 M: 3 F	Zoster, zoster opthalmicus (1 case)	7	5	2	8 d (range, 7–20 d)	10 d post-oral Rx (6 cases)	Valacyclovir (6 patients); hospitalization and iv valacyclovir followed by oral (1 case)
Furer; Israel [33]	Case series	4	BNT16b2 (6); 6 F	Zoster; zoster opthalmicus (1 case)	6	5	1	2.5 d (range, 2–10 d)	4.5 wks (range, 1.5–6 wks)	Acyclovir (2 patients), valacyclovir (2 patients)
Rodríguez-Jiménez; Spain [34]	Case series	4	BNT16b2 (5); 2 M: 3 F	Zoster	5	3	2	3 d (range, 1–16 d)	NR	NR
Chiu; Taiwan [35]	Case series	4	mRNA-1273 (1) AZD1222 (2); 3 M	Zoster	3	3	0	2 d (mRNA-1273); 2 d, 7 d (AZD1222)	1 wk post-oral Rx	Acyclovir
Alpalhão; Portugal [36]	Case series	4	BNT16b2 (2) AZD1222 (2); 1 M: 3 F	Zoster	4	4	0 (dose 2 in 2 patients)	3.5 d (range, 3–6 d)	NR	Valacyclovir
Van Dam; Netherlands [37]	Case series	4	BNT16b2 (2); 1 M: 1 F	Zoster	2	2	0	**Case 1**: 15 d **Case 2**: 13 d	Case 1: 2 wks Case 2: 10 d	Case 1: no Rx Case 2: valacyclovir
Santovito; Switzerland [38]	Case report	5	BNT16b2 (1); 1 M	Zoster	1	0	1	3 d	30 d (Rx started at 72 h)	Prednisone, hydroxyzine, and 2% mupirocin ineffective
David; USA [39]	Case report	5	mRNA-1273 (1); 1 F	Zoster	1	1	NR	2–3 d	NR	None
	**Pityriasis rosea-like eruption**									
Temiz; Turkey [40]	Case series	4	BNT16b2 (14); 4 M: 10 F	Pityriasis rosea-like rash	14	10 BNT16b2	4 BNT16b2	**Dose 1:** 14 d (range, 5–21 d) **Dose 2:** 9 d (range, 4–13 d)	**Dose 1:** 8.5 wks (range, 6–12 wks) **Dose 2:** 4 wks (range, 3–6 wks)	Topical CS, OAH
Català; Spain [2]	Cross-sectional national study	3	BNT16b2 (163), mRNA-1273 (147), AZD1222 (95); 80 M: 325 F	Pityriasis rosea-like rash	20 (5 mRNA-1273, 11 BNT16b2, 4 AZD1222)	12	8	6.3 d (mean)	25.2 (mean)	13 patients treated with TCS, OCS, OAH, analgesics
McMahon; USA [1]	Registry-based study	4	mRNA-1273 (343), BNT16b2 (71); 40 M: 374 F	Pityriasis rosea-like rash	4 (3 BNT16b2)	3 (2 BNT16b2)	1; recurrent (BNT16b2)	**Dose 1:** 14 d **Dose 2:** 4 d	**Dose 1:** 10 d **Dose 2:** 5 d	NR
Niebel; Germany [23]	Case series	4	BNT16b2 (1), AZD1222 (1); 1 M: 1 F	Pityriasis rosea-like rash	2	1 (AZD1222)	1 (BNT16b2)	**Dose 1:** 8 d **Dose 2:** 22 d	NR	TCS, emollients
Cyrenne; Canada [41]	Case series	4	BNT16b2 (2), 1 M: 1 F	Pityriasis rosea-like rash	3	1	2 (1 recurrent)	**Dose 1:** 2 d **Dose 2:** 3 wks	**Dose 1:** 2 wks **Dose 2:** 3 wks	TCS, doxycycline, OAH
Choi, Singapore [16]; Tihy, Switzerland [17]; Yu; Philippines [21]; Cohen; USA [42]; Adya; India [43]; Dormann; Germany [44]; Busto-Leis; Spain [45]; Carbadillo Vazquez; Portugal [46]; Leerunyakal; Thailand [47]; Abdullah; Lebanon [48]; Bostan; Turkey [49]; Larson, USA [50]	Case reports/series	4 [45], 5	BNT16b2 (8), mRNA-1273 (1), AZD1222 (4); 5 M: 8 F	Pityriasis rosea-like rash	14 (9 BNT16b2, 4 AZD1222, 1 mRNA-1273)	8 (3 BNT16b2, 4 AZD1222, 1 mRNA-1273)	6 (BNT16b2); 1 recurrent	**Dose 1**: 4 d (range, 4–14 d) **Dose 2:** 7 d (range, 1–15 d)	1–4 wks	TCS, OAH, symptomatic Rx
	**Pernio, chilblains and purpuric reactions**									
Català; Spain [2]	Cross-sectional national study	3	BNT16b2 (163), mRNA-1273 (147), AZD1222 (95); 80 M: 325 F	Purpuric rash	16 (9 AZD1222, 7 BNT16b2)	11	5	7.6 d	15.7 d	8 patients treated: TCS, OCS, OAH, paracetamol
McMahon; USA [1]	Registry-based study	4	mRNA-1273 (343), BNT16b2 (71); 40 M: 374 F	Pernio/chilblains, petechiae	8 pernio/chilblains (5 BNT16b2), 4 petechiae (3 mRNA-1273)	6 pernio/chilblains (3 BNT16b2), 2 petechiae (1 mRNA-1273)	2 pernio/chilblains (BNT16b2), 2 petechiae (1 mRNA-1273)	**Dose 1:** 10 d (pernio/chilblains), 2 d (patechaie) **Dose 2:** 11 d (pernio/chilblains), 1 d (petechiae)	**Dose 1:** 10.5 d (pernio/chilblains), 4.5 d (petechiae) **Dose 2:** 4.5 d (pernio/chilblains), 3 d (petechiae)	NR
Mazzatenta; Italy [51]	Case series	4	BNT16b2 (3); 1 M: 2 F	Purpuric lesions	3	1	2	**Dose 1:** 10 d **Dose 2:** mean of 22 d	**Dose 1:** 12 d **Dose 2:** mean of 13 d	Self-resolved
Holmes; USA [24]	Case series	4	BNT16b2 (1), mRNA-1273 (1); 2 F	Purpuric, reticulated patches (BNT16b2); chilblain-like papules	2	1 (mRNA-1273)	1 (BNT16b2)	**Dose 1**: 10 d **Dose 2**: 5 d	**Dose 1**: ≤13 d **Dose 2**: ≤2 wks	Case 1: OCS, TCS, OAH Case 2: OAH, TCS
Lopez; USA [52]	Case report	5	BNT16b2; 1 M	Pernio	1	0	1	3 d	28 d	Clobetasol, avoidance of cold
Kha; USA [53]	Case report	5	mRNA-1273; 1 F	Chilblains	1	1	0	2 d	7 d	Clobetasol
Qiao; USA [54]	Case report	5	BNT16b2; 1 F	Chilblains	2	1	1; worsening	**Dose 1**: 2 wks **Dose 2**: h	21 d	Improved with TCS x 2 wks; then worse with rituximab
Annabi; France [27]	Case series	5	BNT16b2; 1 M	Chilblains	1	0	1	5 d	≤7 d	None
Cazzato; Italy [55]	Case report	5	BNT16b2; 1 M	Purpuric rash	1	0	1	15 d	NR	i.v. methylprednisolone
Irvine; USA [56]	Case report	5	BNT16b2; 1 F	Petechiae, desquamation	1	0	1	5 d	3 wks	Monitoring complete blood cell count
Niebel; Germany [23]	Case series	5	BNT16b2; 1 M	Petechial annular plaques	1	0	1	2 d	NR	Dapsone, prednisolone
	**DIR to dermal HA filler**									
McMahon; USA [1]	Registry-based study	4	mRNA-1273 (343), BNT16b2 (71); 40 M: 374 F	DIR to dermal HA filler	9 (1 BNT16b2, 8 mRNA-1273)	3 (mRNA-1273)	6 (5 mRNA-1273)	NR	NR	NR
Munavalli; USA [57]	Case series	4	BNT16b2 (1), mRNA-1273 (1); 2 F	DIR to dermal HA filler	2	1 (mRNA-1273)	1 (BNT16b2)	**Dose 1**: 12 h **Dose 2**: 24 h	**Dose 1** (mRNA-1273): 48 h (lisinopril at 48 h) **Dose 2** (BNT16b2): 24 h	BNT16b2: OCS mRNA-1273: OAH, acetaminophen, lisinopril
Munavalli; USA [58]	Case series	4	mRNA-1273 (2), BNT16b2 (2); 4 F	DIR to dermal HA filler	5 (3 mRNA-1273)	2 (1 BNT16b2)	3 (2 mRNA-1273); 1 recurrent (mRNA-1273)	**Dose 1:** 10 d (BNT16b2), 18 h (mRNA-1273) **Dose 2:** 2 d (BNT16b2), 24h (mRNA-1273)	**Dose 1** (BNT16b2): 7 d **Dose 2** ((BNT16b2): 4 d (lisinopril started at 72 h) **Dose 1** (mRNA-1273): 48 h (lisinopril started at 24 h) **Dose 2** (mRNA-1273): 5 d (lisinopril started at 48 h)	Low-dose lisinopril
Michon; Canada [59]	Case series	4	BNT16b2 (2); 2 F	DIR to dermal HA filler	2	2	NR	**Case 1:** 2 d **Case 2:** a few d	**Case 1:** 5 d **Case 2:** hyaluronidase injected at 3 wks	**Case 1:** no R× **Case 2:** resolved within 48 h after hyaluronidase injection
	**Unusual reactions**									
Català; Spain [2]	Cross-sectional national study	3	BNT16b2 (163), mRNA-1273 (147), AZD1222 (95); 80 M: 325 F	Papulovesicular	26 (7 mRNA-1273, 11 BNT16b2, 8 AZD1222)	18	8	6.4 d (mean)	19.3 d (mean)	OAH, TCS, SCS, topical antibiotics, paracetamol
McMahon; USA [1]	Registry-based study	4	mRNA-1273 (343), BNT16b2 (71); 40 M: 374 F	Vesicular	10 (5 mRNA-1273)	7 (4 mRNA-1273)	3 (2 BNT16b2)	**Dose 1:** 7 d **Dose 2:** 3 d	**Dose 1:** 7 d **Dose 2:** 7 d	NR
Niebel; Germany [23]	Case series	4	BNT16b2 (2); 2 F	Vesicular	2	0	2	3 d, 7 d	NR	TCS, topical fusidine
Coto-Segura; Spain [60]	Case series	4	BNT16b2 (4); 4 M	Bullous pemphigoid (3); linear IgA dermatosis (1)	4	4	NR	3–17 d	NR	NR
Larson; USA [50]	Case series	4	BNT16b2 (1), mRNA-1273 (1); 2 M	New-onset bullous pemphigoid	2	1 (BNT16b2)	1	**Dose 1:** 3 wks **Dose 2:** 2 wks	**NR**	Improvement with OCS, TCS, doxycycline, niacinamide, OAH
McMahon; USA [1]	Registry-based study	3	mRNA-1273 (343), BNT16b2 (71); 40 M: 374 F	Erythromelalgia	14 (11 mRNA-1273, 3 BNT16b2)	6 (5 mRNA-1273)	8 (6 mRNA-1273)	**Dose 1:** 7 d **Dose 2:** 1 d	**Dose 1:** 5.5 d **Dose 2:** 3 d	NR
Leasure; USA [61]	Case series	4	BNT16b2 (2); 1 M: 1 F	Generalized eczematous eruption	4	2	2; recurrent	**Dose 1:** 5 d (mean) **Dose 2:** 9 d (mean)	Several wks	TCS, OAH, OCS
Corbeddu; Italy [22]	Case series	5	BNT16b2; 1 M	Generalized eczematous eruption	1	1	NR	2 d	2–3 d	OCS
Holmes; USA [24]	Case series	5	BNT16b2; 1 M	Eczematous dermatitis	2	1	1; recurrent	**Dose 1:** ≤7 d **Dose 2**: NR	2–3 wks	TCS, tacrolimus ointment
Niebel; Germany [23]	Case series	4 5	BNT16b2 (3); 2 M: 1 F BNT16b2; 1 M	Hematogenous contact dermatitis Psoriasiform flare of atopic dermatitis	3 1	2 1	1 0	**Dose1**: 2 d, 3 d **Dose 2**: 12 d 21 d	NR NR	TCS; prednisolone and cyclosporine (1 case) Prednisolone, TCS, narrowband UVB
McMahon; USA [1]	Registry-based study	4	mRNA-1273 (343), BNT16b2 (71); 40 M: 374 F	Erythema multiforme	3 (mRNA-1273)	3	0	N/A	N/A	NR
Gambichler; Germany [62] Lavery; UK [63]	Case report Case report	5 5	BNT16b2; 1 F BNT16b2; 1 F	Erythema multiforme, Rowell syndrome Erythema multiforme flare	1 2	1 1	NR 1; recurrent	1 d **Dose 1**: 12 h **Dose 2**: 24 h	NR NR	OCS TCS
Tihy; Switzerland [17]	Case series	5	BNT16b2; 1 F	Prurigo nodularis	1	0	1	15 d	2 wks	NR
Soyfer; Israel [64]	Case series	4	BNT16b2 (2); 2 M	Radiation recall dermatitis	2	0	2	6 d	≤7 d	TCS, analgesics
Lim; UK [65]	Case report	5	AZD1222; 1 M	SDRIFE-like eruption	1	NR	1	1 d	1 mo	OCS, potassium permanganate soaks betamethasone/clotrimazole,
Dash; India [66]	Case report	5	AZD1222; 1 M	SJS	1	1	No dose 2	3 d	14 d	Cyclosporine
Elboraey; Saudi Arabia [67] Bakir; Saudi Arabia [68]	Case report Case report	5 5	BNT16b2; 1 F BNT16b2; 1 F	SJS TEN	1 1	0 1	1 NR	5 d 7 d	NS 22 d post-etanercept	OCS admission, multidisciplinary care, etanercept
Majid; India [69]	Case report	5	AZD1222; 1 F	Sweet’s syndrome	1	1	NR	7 d	4 wks (betamethasone started at 3 wks)	Injectable betamethasone
Torrealba-Acosta; USA [70]	Case report	5	mRNA-1273; 1 M	Sweet’s syndrome	1	1	NR	1 d	≥13 d	Oral antibiotics, systemic antivirals, SCS
Kaminetsky; USA [71]	Case report	5	mRNA-1273; 1 F	Vitiligo	2	1	1; worsening	Several days	NR	NR
Sandu; India [72] Mücke; Germany [73] Larson; USA [50]	Case series Case report Case series	4 5 4	AZD1222; 1 M: 1 F BNT16b2; 1 M BNT16b2 (1), mRNA-1273 (1); 2 F	Vasculitis Vasculitis Vasculitis	3 1 2	2 0 1 (mRNA-1273)	1; recurrent 1 1 (BNT16b2)	**Dose 1**: 5 d, 7 d **Dose 2**: 2 d 12 d **Dose 1**: day of **Dose 2**: 7 d	**Dose 1**: 2 wks, 7 d **Dose 2**: NR 5 d post-Rx initiation NR	OCS, TCS OCS OAH, SCS, oral antibiotics, dapsone, TCS
Annabi; France [27]	Case series	5 5	BNT16b2; 1 F mRNA-1273; 1 F	Livedo racemose Fixed drug eruption	1 1	1 0	0 1	12 d 2 d	PIH at 2 mos 5 d	None TCS
Hunjun; UK [74]	Case report	5	BNT16b2; 1 M	Pityriasis rubra pilaris-like	2	1	1; worsening	**Dose 1**: 3 d **Dose 2**: a few d	NR	Acitretin, TCS
Merrill; USA [75]	Case series	4	mRNA-1273; 2 M	Facial pustular neutrophilic eruption	2	1	1	≤24 h	≤7 d, >15 d	Case 1: Cephalexin, TCS Case 2: vancomycin, piperacillin/tazobactam, tacrolimus 0.1%, doxycycline
Lopatynsky-Reyes; Costa Rica, Mexico [76]	Case series	4	BNT16b2 (1), mRNA-1273 (1); 2 F	BCG scar local skin inflammation	2	0	2	2 d, 36 h	4 d, 2 d	None
Bostan; Turkey [77] Niebel; Germany [23] Krajewski; Poland [78]	Case report Case series Case report	5 5 5	BNT16b2; 1 M BNT16b2; 1 M BNT16b2; 1 M	Psoriasis vulgaris flare Psoriasis vulgaris flare Psoriasis vulgaris flare	2 1 1	1 0 0	1; recurrent 1 1	**Dose 1**: NS **Dose 2**: 2 wks 20 d 1 d	NR NR NR	NR Cignoline, TCS, narrowband UVB, tildrakizumab NR
Perna; USA [79]	Case report	5	BNT16b2; 1 M	Generalized pustular psoriasis	1	1	NR	5 d	12 d (cyclosporine started at 7 d)	Cyclosporine
Lehmann; Switzerland [80]	Case report	5	BNT16b2; 1 F	New onset guttate psoriasis	2	1	1 (recurrent)	**Dose 1**: 10 d	NS	Clobetasol, ultraviolet light B, betamethasone/calcipotriene
Quattrini; Italy [81]	Case report	5	BNT16b2; 1 F	Palmoplantar psoriasis flare	1	NR	1	48 h	Rapid improvement	OCS, methotrexate
Niebel; Germany [23]	Case series	4	BNT16b2 (2), mRNA-1273 (2); 1 M: 3 F	CLE (3 cases; 2 with mRNA-1273), CLE flare (BNT16b2; 1 case)	4	3	1 (mRNA-1273)	7 d (range, 5–10 d)	3 wks post-OCS Rx	Prednisolone; single cases with OAH, hydroxychloroquine, methotrexate, etoricoxib, TCS
Joseph; USA [82]	Case report	5	mRNA-1273; 1 F	Subacute CLE flare	2	1	1; worsening	Dose 1: 4 d	NR	OCS continuation, Mycophenolate mofetil dose increase, TCS
Elbaek; Denmark [83]	Case report	5	AZD1222; 1 F	Darier’s disease flare	1	1	No dose 2	2 d	>6 wks	TCS, salicylic acid, oral isotretinoin
Hiltun; Spain [84]	Case report	5	BNT16b2; 1 F	Lichen planus flare	1	0	1	48 h	NR	TCS

BCG, bacillus Calmette-Guérin; CLE, cutaneous lupus erythematosus; D, day(s); DIR, delayed inflammatory reaction; DLLR, delayed large local reaction; F, female(s); FDA, Food and Drug Administration; h, hour/hours; HA, hyaluronic acid; HSV, herpes simplex virus; i.m., intramuscular; i.v., intravenous; LFTs, liver function tests; M, male(s); NR, not reported; NSAIDS, non-steroidal anti-inflammatory drugs; OAH, oral antihistamine; OCS, oral corticosteroid; PIH, post-inflammatory hyperpigmentation; Rx, treatment; SDRIFE, symmetrical, drug-related intertriginous and flexural erythema; SCS, systemic corticosteroid; SJS, Stevens Johnson Syndrome; TCS; topical corticosteroid; Team CC-R, CDC COVID-19 Response Team; TEN, toxic epidermal necrolysis; VZV, varicella zoster virus; wks, weeks. * Rating score of the studies was ranked according to Quality Rating Scheme for Studies and Other Evidence [5] and Oxford Centre for Evidence-based Medicine for ratings of individual studies [6].

**Table 2 microorganisms-10-00624-t002:** Cutaneous reactions by COVID-19 vaccine type.

Reaction	mRNA-1273 No. (%) ^a^	BNT162b2 No. (%)	ADZ1222 No. (%)	Row Total ^b^
Delayed large local reaction (‘COVID arm’)	411 (72)	140 (24.5)	20 (3.5)	571 (40.4)
Local injection site reaction	186 (79.5)	48 (20.5)	0	234 (16.5)
Urticaria	47 (36.7)	57 (44.5)	24 (18.8)	128 (9.0)
Morbilliform/diffuse erythematous eruption	31 (33.0)	53 (55.3)	11 (11.7)	95 (6.7)
VZV/HSV reactivation	48 (29.8)	90 (59.6)	17 (10.6)	155 (11.0) ^c^
Pityriasis rosea-like eruption	7 (12.3)	41 (71.9)	9 (15.8)	57 (4.0)
Pernio, chilblains, and purpuric eruptions	7 (17.1)	25 (61.0)	9 (22.0)	41 (2.9)
DIR to dermal HA filler	12 (66.7)	6 (33.3)	0	18 (1.3)
Unusual reactions	39 (33.6)	62 (53.4)	15 (12.9)	116 (8.2)
Column total	788 (55.7)	522 (36.9)	105 (7.4)	1415

DIR, delayed inflammatory reaction; HA, hyaluronic acid; HSV, herpes simplex virus; VZV, varicella zoster virus. ^a^ Row percentages are provided, rounded to first decimal. ^b^ Column percentages are provided in parentheses. ^c^ Zoster accounts for 135 of 155 reactions and 9.5% of total reactions.

**Table 3 microorganisms-10-00624-t003:** Suggested pathogenetic mechanisms underlying COVID-19 vaccine-related cutaneous reactions.

Skin Reaction	Potential Mechanisms
Delayed large local reaction	T-cell mediated responses to a vaccine excipient, lipid nanoparticle, or mRNA component [10,11]
Urticaria	IgE-mediated reactions are more typically associated with the inactive components of the vaccine (i.e., egg proteins, gelatin, and latex) [85]
Anaphylaxis	Pre-existing antibody recognition of the vaccine excipient polyethylene glycol (PEG); contact system activation by nucleic acid; complement recognition of the vaccine-activating allergic effector cells; direct mast cell activation [85]
Morbilliform eruption	Immune activation-mediated skin response; prior coronavirus infection may generate a cross-reaction with antigen that mRNA vaccine encodes [86]
VZV/HSV reactivation	Innate or cell-mediated immune defense failures initiated by the host in response to mRNA COVID-19 vaccines; [22] strong immune response against the S protein from vaccine may distract the cell-mediated control of another, latent virus [2]
Pityriasis rosea-like eruption	Vaccination leads to a state of altered immunity and may lead to endogenous reactivation of HHV-6 or HHV-7 [40]; T-cell mediated response triggered by molecular mimicry from a viral epitope [41]
Pernio, chilblains, and purpuric lesions	Vaccine-induced microangiopathy [51]; viral proteins in the endothelial cells of the dermal vessels and accumulation of immune complexes that activate the complement cascade, causing small vessel wall damage [52]
DIR to dermal hyaluronic acid fillers	COVID-19 spike protein interacts with ACE2 receptors which trigger pro-inflammatory loco-regional TH1 cascade and promote a CD8 and T cell mediated reaction to incipient granulomas [57,58]
Vesiculobullous lesions	Cross-reactions between SARS-CoV-2 spike protein antibody and tissue proteins such as transglutaminase 2 and 3, collagen, and S100B antigen may play a role in developing these immune-mediated skin lesions [60]
Generalized eczematous eruptions	Vaccine may act as an environmental trigger in a genetically susceptible individual (i.e., personal/family history of atopy) [61]
Radiation recall dermatitis	Offending agent upregulates inflammatory cytokines that are already increased in area of irradiation, leading to a local hypersensitivity reaction [87]
SDRIFE-like eruption	Co-infection by other viruses or uncommon clinical presentation of post-vaccination hyperviscosity [88]
Stevens-Johnson syndrome	Expression of vaccine antigens on keratinocytes leads to a CD8+ T-cell response against epidermal cells, thus causing apoptosis of keratinocytes and detachment of dermo-epidermal junction in a genetically susceptible individual [66]
Psoriasis exacerbation	Vaccine increases IL-6 production and recruitment of Th17 cells which are involved in psoriasis; [78] vaccine may activate the plasmacytoid and dermal myeloid dendritic cells, which upregulate type I IFNs that initiate the inflammatory cascade; [81] mRNA vaccines bind to Toll-like receptors that result in increased production of type I IFNs [45]

ACE2 angiotensin-converting enzyme 2; HHV, human herpes virus; HSV, herpes simplex virus; IFNs, interferons; mRNA, messenger RNA; SDRIFE, symmetrical drug-related intertriginous and flexural exanthem; Th1, T helper 1; VZV, varicella zoster virus.

## Data Availability

Data are available on request from the authors.

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
