# Peer review of "Cutaneous Complications of mRNA and AZD1222 COVID-19 Vaccines: A Worldwide Review"

_microorganisms, 2022, doi:10.3390/microorganisms10030624_

Round 1
Reviewer 1 Report
The authors aimed to describe cutaneous reactions related to 3 different covid-19 vaccines. They decided to use a systematic review approach and not a narrative way to present the evidence and they nicely presented the potential underlying mechanism for different reactions. Knowing what type of reactions and when should be expected to occur is clinically relevant and useful, however a more interesting question is how common these are among those vaccinated e.g what is the % of subjects developing cutaneous reactions after covid-19 vaccination? Most likely there are studies, including RCTs and cohort studies, reporting that and this information could add value to this work. Overall, as a systematic review this work seems to have some flaws in its methodology and reporting of results, therefore clarifications and extra information may be necessary:
SR Methodology
- Please clarify the following:
- whether a systematic review protocol (defining PICO/PECO) was developed and if yes whether this was published in PROSPERO or elsewhere prior to data extraction.
- whether you followed the PRISMA guidance (see Page MJ, McKenzie JE, Bossuyt PM, Boutron I, Hoffmann TC, Mulrow CD, et al. The PRISMA 2020 statement: an updated guideline for reporting systematic reviews. BMJ 2021;372:n71. doi: 10.1136/bmj.n71 . For more information, visit: http://www.prisma-statement.org/
- Please elaborate on the assessment of risk of bias (RoB) of included studies. Did the authors use a validated tool for RoB assessment e.g. Cochrane ROBIN or Risk of Bias (RoB2) for non-randomized and randomized studies respectively, or other tools for observational studies e.g. Newcastle–Ottawa Scale [NOS]?
- Why did the authors exclude studies with incomplete data on cutaneous reaction? If studies assess cutaneous reactions most likely they were eligible and could be included. For that, the authors should assess the risk of bias as high or unclear if the data or information on cutaneous reactions was not provided in a proper manner (e.g. selective reporting bias). Usually, incomplete reporting of information is not an exclusion criterion for SR but a bias. Similarly, the authors excluded studies with incomplete clinical data (e.g., clinical features and/or course of the eruption). You may have missed important large observational (real-life) studies reporting reactions but not their course. That could add to the number of reactions and change your results and conclusions dramatically.
- It is also unclear why studies with self-reported reactions where excluded. Unless there is a good justification for excluding studies with self-reported cutaneous reactions, those should be included and be assess with the respective risk of bias on the method of outcome assessment so you.
Results
- I was surprised that large observational prospective studies and vaccine RCTs were not included after screening. Can the authors explain the reason behind that gab?
A few examples missed among others are provided below:
Robinson LB, Fu X, Hashimoto D, et al. Incidence of Cutaneous Reactions After Messenger RNA COVID-19 Vaccines. JAMA Dermatol. 2021;157(8):1000–1002. doi:10.1001/jamadermatol.2021.2114.
N Engl J Med 2020; 383:2603-2615 DOI: 10.1056/NEJMoa2034577.
Grieco T, et al Cutaneous adverse reactions after COVID-19 vaccines in a cohort of 2740 Italian subjects: An observational study. Dermatol Ther. 2021 Nov;34(6):e15153. doi: 10.1111/dth.15153. Epub 2021 Oct 13. PMID: 34622531; PMCID: PMC8646410.
Alhumaid, S., et al. Anaphylactic and nonanaphylactic reactions to SARS-CoV-2 vaccines: a systematic review and meta-analysis. Allergy Asthma Clin Immunol 17, 109 (2021). https://doi.org/10.1186/s13223-021-00613-7
- Age of participants, history of allergy or anaphylaxis and duration of follow up for adverse events are important and relevant information which the authors may consider including if available in individual studies.
- The authors do not report their findings on risk of bias assessment. What is the number of studies with low/high risk of bias? Please include a table or figure for RoB.
- It would have been more useful to have a first table with the characteristics of included studies (e.g., author, publication year, country, study design, number of participants, ethnic groups, special or general population, type of vaccine, method of outcome assessment, duration of follow up, mean age, gender, and Risk of bias of each study.
- A second separate table for the outcome (cutaneous reactions) including the % of reactions reported in each study (n/N and %) if available may provide a more complete picture of reactions. If the authors decide to include the incidence of cutaneous reactions, a meta-analysis /forest plot should be included if the number of studies allows it.
- Could differences in type of reactions exist in different age groups?
- On your PRISMA flowchart you may report the total number of results before removing duplicates
Discussion
- Your conclusion focuses on the role of dermatologists and other specialists, however in most countries, it is the primary care physician who is the first contact of patients with healthcare. Therefore, the role of GPs should not be minimized or omitted.
Reviewer 2 Report
The authors present a very interesting review article about the COVID-19 vaccination cutaneous complications. This article is clear and well written. The study selection and extraction of data are correct and results are clear. I particularly appreciated the table 3 where the authors present the suggested pathogenetic mechanism underlying the COVID -19 vaccine – related cutaneous reactions. I have just a remark: in table 2 the authors present 2 cases of bullous pemphigoid COVID-19 vaccine related referring only one reference (51), but in the text (page 19 line 244) the authors report “ some cases showed features of bullous pemphigoid or linear IgA dermatosis” citing 3 references (1,51,62). The authors should control in a better way the real incidence of autoimmune bullous disease induced or worsened by COVID vaccine.
Accepted with minor revision
Round 2
Reviewer 1 Report
I would like to thank the authors for their response. They authors claim that they feel that their systematic approach has been complete however they seem to lack two important principles of a comprehensive systematic review: a) a registered protocol and b) selective reporting of outcomes bias (Justification is provided below).
It is recognisable that case reports/series may be particularly important when conducting a systematic review of a body of evidence that consists primarily of uncontrolled clinical observations. Even though this is not totally true for adverse effects of such a widely used intervention worldwide as such covid-19 vaccines, If the authors consider that only small case reports and case series studies can addressed the main objectives of their study, then it is preferred to use a systematic approach appropriate for this kind of studies, or to summarize the evidence as a narrative review; but a quantitative synthesis is susceptible to bias while missing larger studies of higher quality and may draw misleading conclusions. Please allow me to doubt whether in in real-life covid arm is twice more common than local reaction as your data indicate.
Publishing a systematic review protocol as a peer-reviewed article has the advantage that independent reviewers will critically appraise the proposed methods (PLoS Med. 2011, doi: 10.1371/journal.pmed.1001009). Therefore, it would have provided the readers and reviewers with a transparent of the original PICO/PECO and aim(s) of this study, which as stated by the authors that was: The objective of this systematic review was to assess the dermatologic complications of mRNA-1273 (Moderna; mRNA vaccine), BNT162b2, and AZD1222 (AstraZeneca-41 Oxford University; adenovirus vector vaccine) vaccination.
Although it would have been important to see updates of existing SR on incidence of cutaneous reactions, I respect that the authors chose not to include that objective. But the aforementioned objective, by definition, should have included adverse effects reported by clinical trials and larger observational studies, so one may wonder why this search strategy (as provided) have not capture more of these studies to be included. Case reports and case series studies are de novo observational studies high risk for bias and it is not common to plan a SR which only targets this type of studies. To be more precise most SR exclude this type of studies in their protocols.
Moreover, in their response the authors state that they aimed to specifically describe the morphology, history, timing and duration, treatment (Table 1), distribution of reactions per vaccine type (Table 2), and mechanisms (Table 3) of the reactions in this review – we do not merely perform a study of prevalence. This specific objective differs from the one stated in the manuscript, and certainly it would require a more detailed search strategy for different PICO inclusion/exclusion criteria for the SR.
Furthermore, the authors responded that studies with self-reported reactions were excluded because the diagnosis was not made by a health care provider and could have been wrong, and the rash was not examined by a health care provider. Without a registered protocol clearly providing the outcomes of the study, it is not obvious that this decision had been made and justified prior to data collection. When we are assessing a widely used intervention comprising millions of people, driving conclusions on specific and rate side effects only by small case reports and case series underlies risk of biased conclusions.
Nevertheless, in high quality, comprehensive SRs self-reported outcomes usually are included but assessed as high risk for outcome assessment if necessary. The Cochrane states the following key points on the matter:
- Summary data on patient-reported outcomes (PROs) are important to ensure healthcare decision makers are informed about the outcomes most meaningful to patients.
- Authors of systematic reviews that include PROs should have a good understanding of how patient-reported outcome measures (PROMs) are developed, including the constructs they are intended to measure, their reliability, validity and responsiveness.
- Authors should pre-specify at the protocol stage a hierarchy of preferred PROMs to measure the outcomes of interest.
Johnston BC, Patrick DL, Devji T, Maxwell LJ, Bingham III CO, Beaton D, Boers M, Briel M, Busse JW, Carrasco-Labra A, Christensen R, da Costa BR, El Dib R, Lyddiatt A, Ostelo RW, Shea B, Singh J, Terwee CB, Williamson PR, Gagnier JJ, Tugwell P, Guyatt GH. Chapter 18: Patient-reported outcomes. In: Higgins JPT, Thomas J, Chandler J, Cumpston M, Li T, Page MJ, Welch VA (editors). Cochrane Handbook for Systematic Reviews of Interventions version 6.2 (updated February 2021). Cochrane, 2021. Available from www.training.cochrane.org/handbook.
I understand the authors’ eager to present details on the type and natural history of reactions and their underlying mechanisms and I also value their clinical relevance, however the quantitative part of this review is susceptible to bias without including larger and higher quality studies.
I would like to thank the authors for trying to address the risk of bias of included studies, instead of omitting completely an important component of a SR. As mentioned earlier the type of studies included in this review are susceptible to bias. The authors instead of speculating regarding selection and information bias overall, may use a more systematic approach, as the one proposed by Murad et al (or similar) on methodological quality and synthesis of case series and case reports, which provides a framework to evaluate the methodological quality of case reports/series and synthesise their results (BMJ Evid Based Med. 2018 Apr; 23(2): 60–63. doi: 10.1136/bmjebm-2017-110853.) Other available tools include JBI critical appraisal checklist https://jbi.global/critical-appraisal-tools (Military Med Res 7, 7 (2020). https://doi.org/10.1186/s40779-020-00238-8)
Finally, the response on the reasoning for excluding studies with incomplete data on cutaneous outcomes/adverse events is not satisfactory as they have neglected the bias from selective reporting of outcome in eligible studies. In many studies, a range of outcome measures is recorded but not all are reported (Pocock 1987, Tannock 1996). The choice of outcomes that are reported can be influenced by the results, potentially making published results misleading. According to Cochrane the assessment of risk of bias due to selective reporting of outcomes should be made for the study as a whole, rather than for each outcome. The study-level judgement provides an assessment of the overall susceptibility of the study to selective reporting bias. https://handbook-5-1.cochrane.org/chapter_8/8_14_2_assessing_risk_of_bias_from_selective_reporting_of.htm & https://methods.cochrane.org/bias/reporting-biases )